# Air Passenger Travel and International Surveillance Data Predict Spatiotemporal Variation in Measles Importations to the United States

**DOI:** 10.3390/pathogens10020155

**Published:** 2021-02-03

**Authors:** Marya L. Poterek, Moritz U. G. Kraemer, Alexander Watts, Kamran Khan, T. Alex Perkins

**Affiliations:** 1Department of Biological Sciences and Eck Institute for Global Health, University of Notre Dame, Notre Dame, IN 46556, USA; 2Department of Zoology, University of Oxford, Oxford OX1 3SY, UK; moritz.kraemer@zoo.ox.ac.uk; 3Li Ka Shing Knowledge Institute, St. Michael’s Hospital, Toronto, ON M5B 1T8, Canada; Alexander@bluedot.global (A.W.); Khank@smh.ca (K.K.); 4BlueDot, Toronto, ON M5J 1A7, Canada; 5Department of Medicine, University of Toronto, Toronto, ON M5S 3H2, Canada

**Keywords:** air travel, importation, global health, measles

## Abstract

Measles incidence in the United States has grown dramatically, as vaccination rates are declining and transmission internationally is on the rise. Because imported cases are necessary drivers of outbreaks in non-endemic settings, predicting measles outbreaks in the US depends on predicting imported cases. To assess the predictability of imported measles cases, we performed a regression of imported measles cases in the US against an inflow variable that combines air travel data with international measles surveillance data. To understand the contribution of each data type to these predictions, we repeated the regression analysis with alternative versions of the inflow variable that replaced each data type with averaged values and with versions of the inflow variable that used modeled inputs. We assessed the performance of these regression models using correlation, coverage probability, and area under the curve statistics, including with resampling and cross-validation. Our regression model had good predictive ability with respect to the presence or absence of imported cases in a given state in a given year (area under the curve of the receiver operating characteristic curve (AUC) = 0.78) and the magnitude of imported cases (Pearson correlation = 0.84). By comparing alternative versions of the inflow variable averaging over different inputs, we found that both air travel data and international surveillance data contribute to the model’s ability to predict numbers of imported cases and individually contribute to its ability to predict the presence or absence of imported cases. Predicted sources of imported measles cases varied considerably across years and US states, depending on which countries had high measles activity in a given year. Our results emphasize the importance of the relationship between global connectedness and the spread of measles. This study provides a framework for predicting and understanding imported case dynamics that could inform future studies and outbreak prevention efforts.

## 1. Introduction

Despite the elimination of endemic measles in the United States in the 1990s, the number of measles cases and outbreaks has begun to rise in recent years, as measles–mumps–rubella (MMR) vaccination rates in the US are declining and transmission internationally is on the rise [1]. Measles virus is highly infectious and can cause serious symptoms and even death, and cases imported to the US can result in outbreaks that harm vulnerable populations, particularly now as herd immunity wanes under the influence of anti-vaccine sentiment [2]. Recent outbreaks have required significant public health responses and have led to stricter vaccination policies in some states [3].

Measles outbreaks often occur when infectious individuals enter into largely unvaccinated populations. This was the case in the 2019 New York outbreak, where several infected travelers carried measles into orthodox Jewish neighborhoods with low school immunization rates, resulting in hundreds of epidemiologically linked cases [4]. Even a single infected traveler can spark a significant outbreak—a recent 71-case outbreak near Portland, Oregon was the result of a single international importation into an area with vaccination rates significantly lower than those required to maintain herd immunity [3,5]. Travelers such as these, carrying the measles virus internationally, pose a risk to susceptible individuals both while in transit and at their final destination and are often the primary drivers of measles dynamics in non-endemic countries [6]. 

Many studies have acknowledged the importance of importation in both pre- and post-elimination era US measles outbreaks [7,8]. Imported cases are the result of measles transmission outside of the US, regardless of an infected individual’s nationality, and are inevitable while endemic measles remains present in many parts of the world [9]. Ultimately, these importations are driven by a combination of incidence in source countries and international travel patterns, suggesting that importation into the US will continue until elimination is achieved globally [10].

Patterns of imported cases have been modeled for many other diseases, including MERS, chikungunya, dengue, Zika, Ebola, Lassa fever, and COVID-19, and have focused on the relationship between air travel and importation risk [11,12,13,14,15,16,17,18]. A previous implementation of travel-driven measles importation into New Zealand used travel data and global measles incidence data to determine age-specific importation risk with a generalized linear model [19]. While existing studies have explored the relationship between international measles incidence, travel, and indices of importation risk, an understanding of how this relationship translates into actual imported case count predictions remains an outstanding question.

In this study, we leveraged publicly available data from the World Health Organization (WHO) and the US Centers for Disease Control and Transmission (CDC) as well as proprietary flight data to fit a single-parameter regression model across the years 2011–2016. We used this model to make geographically specific predictions about the volume and likely origins of imported measles cases into the US and incorporated reported imported cases that made validation of our predictions possible.

Analysis of averaged values as modeled inputs allowed us to isolate the influence of passenger volume and international measles incidence on predictive performance; we additionally considered a null scenario in which state populations were substituted as input to the model. The inclusion of a variety of data types in this framework allowed us to model and explain spatial and temporal variation in measles importation patterns in the US. Further exploration of potential origins of imported measles cases made it possible to consider the utility of predictive modeling techniques to targeted disease surveillance and mitigation efforts.

## 2. Results

### 2.1. Inflow Variable Informed by Air Travel and International Surveillance Data

We found that the inflow variable with full detail about air travel data and international surveillance data was associated with a high correlation between predicted and observed imported cases (0.84), which increased slightly when state-year combinations with zero imported cases were omitted (0.85) (Table 1). Likewise, we found that the 95% prediction interval of this inflow variable had a coverage probability of 95%, with data falling outside the prediction interval limited to state-year combinations with six or fewer imported cases (Figure 1A). With respect to predicting the presence of imported cases in a given state and year, the inflow variable with full detail had an area under the curve of the receiver operating characteristic curve (AUC) of 0.78. A similar AUC of 0.78 ± 0.06 (SD) was obtained through repetitive resampling and cross-validation, suggesting that predictions of the presence of imported cases were significantly better than at random (Table 2, Appendix A).

For reference, a model based on state population alone had a correlation of 0.66 and an AUC of 0.58 when fitted to the full data set (Table 1), and similar values in cross-validation (Table 2). Thus, while there was some predictive value associated with inflow based on state population alone, the addition of air travel and international surveillance data to the inflow variable resulted in a marked improvement in the ability to predict the presence and number of imported cases across states and years.

### 2.2. Comparison with Averaged Inflow Variables

For the inflow variables that averaged over either air travel or international surveillance data, correlation was the statistic for which there was the greatest difference with the inflow variable with full detail. For both of these averaged inflow variables, correlation dropped from 0.84 to 0.76 (Table 1). This tendency held in cross-validation as well, with correlation dropping from 0.78 in the case of the inflow variable with full detail to 0.71 ± 0.2 for the averaged inflow variables (Table 2). In contrast, values of AUC associated with all three of these inflow variables were very similar (Table 1), including in cross-validation (Table 2). In terms of coverage of the prediction interval, the inflow variable with full detail captured 7/7 state-year combinations with seven or more imported cases (Figure 1A), and inflow variables that averaged over one or the other data type both accounted for 6/7 (Figure 1B,C). Thus, in some respects, there was a decrease in predictive performance when one data type was averaged, but retaining either data type still enabled reasonably good prediction of imported cases based on inflow.

### 2.3. Comparison with Modeled Inflow Variables

Relative to the inflow variable with full detail, the inflow variable that used modeled international surveillance showed a drop in predictive performance. Its overall correlation was 0.77, its AUC was 0.79, and similar values were obtained in cross-validation (Table 1 and Table 2). Like the inflow variable with full detail, however, its prediction interval covered most observations (95%), including those in which the highest numbers of imported cases were observed (Figure 1). Overall, predictions of imported cases based on modeled values of international surveillance performed similar to predictions based on the inflow variable with averaged values of raw international surveillance data, suggesting that the steps taken in generating modeled values negatively impacted the predictive value of international surveillance for the purpose of predicting imported cases.

The inflow variable with modeled flight data performed worst among all the inflow variables that we evaluated. Its AUC was lower than all inflow variables except the null based on state population alone, and its overall correlation was lower than even that null inflow variable (Table 1 and Table 2). Moreover, a much larger proportion of observations (18%) fell outside its 95% prediction interval than under other inflow variables (Figure 1). Overall, predictions of imported cases based on modeled values of air travel were poor, and state population alone may offer an easier and more robust prediction of imported measles cases.

### 2.4. Predicting Measles Origin and Magnitude

Using the inflow variable with full detail, we explored patterns in the predicted magnitude and potential origins of each state’s imported cases from 2011 to 2016 (Figure 2). The expected number of imported cases was generally highest in the most populous states – notably, California, New York/New Jersey, Texas, Florida, and Illinois. In particular, California was predicted to have no fewer than 4.0 (95% prediction interval: 0–9) imported cases in a single year (2012) and as many as 17.0 (95% prediction interval: 8–29) (2014). Other than these six most populous states, most states were predicted to have fewer than two imported cases each year. This did vary by year, however, with one to three imported cases predicted for several states in 2011.

The estimated origins of imported cases also varied from year to year, and often correlated with trends in incidence in potential countries of origin (Figure 2). In 2011, imported cases were predicted to have been predominantly from France, which experienced a large outbreak that year [20]. In other years, other countries predominated—notably, the United Kingdom in 2012, the Philippines in 2014, and the Netherlands in 2016. In some cases, the predicted origins of imported measles cases varied among states. For example, in 2015 and 2016, a high proportion of the imported cases to Louisiana, Mississippi, and Texas were predicted to have originated in Equatorial Guinea, but much less so in other states. This resulted from a combination of high measles incidence in Equatorial Guinea in those years together with much higher levels of air travel between Equatorial Guinea and those states. 

### 2.5. Association between Imported and Indigenous Measles Cases

Although our analysis focused on imported cases, we briefly explored the association between imported and indigenous cases to better understand the relevance of our analysis to measles outbreaks in the US. To do so, we calculated the Kendall rank correlation coefficient between imported and indigenous US measles cases combined across all years in 2011–2016 and found a moderately large correlation of 0.45 that was statistically significant (*p* < 2 × 10^−16^). While this suggests that local factors in the US, such as vaccination coverage, play an important role in determining the extent of indigenous measles cases in the US, so too do imported measles cases.

## 3. Discussion

Through regression of imported measles cases against multiple inflow variables, our analysis demonstrated the utility of air travel data and international surveillance data for predicting spatiotemporal variation in imported measles cases in the US. Cross-validation of these predictions verified their robustness, and comparison of predictions based on alternative inflow variables isolated the predictive value of alternative inputs for informing predictions of inflow. Inspection of our model’s predictions with respect to state, year, and likely country of origin showed that predictions of imported cases in a given state and year depended on the state’s population, the countries with which that state was linked via air travel, and contemporaneous measles activity in the countries with which it was most strongly linked.

A unique aspect of our analysis was the use of alternative inflow variables with different combinations of data inputs, which allowed us to isolate the effect of each data type on our predictions. Most of the inflow variables that we considered performed similarly well at predicting the presence or absence of imported measles cases. There was greater discrepancy, however, in how well the inflow variables predicted the numbers of imported measles cases across states and years. Because each regression that we performed involved estimation of only a single parameter, these differences can be directly attributed to the data inputs. Comparison of predictions based on the inflow variable with full detail and those based on inflow variables with averaged inputs showed that air travel data and international surveillance data contributed roughly equally to the success of the model’s predictions. In other words, each data type added some value on its own, but the two data types performed best when paired together.

Substituting either data input for a model-based alternative also reduced the quality of our predictions. In the case of modeled international surveillance, poorer performance could reflect that model-based revisions of measles burden work well for estimating burden within those countries [21] but less well for characterizing measles burden among the subset of the population likely to travel to the US [22,23]. In the case of modeled air travel, possible limitations include omission of airports below a certain size and the inability to pick up on idiosyncratic connections [24], such as the strong connection between Equatorial Guinea and states in the western Gulf Coast of the US—which could be related to the energy industry [25]—that emerged in our exploration of the possible country of origin of imported cases. In the era of COVID-19, air travel patterns are likely to continue to depart from historical trends [26,27], providing further impetus for working with non-modeled air travel data appropriate to the time frame of interest.

Previous studies have identified air travel as a major driver of the importation of measles and a variety of other infectious diseases and have found value in incorporating reported imported case data [12,13,19]. By isolating air travel and incidence in model inputs, we were able to additionally explore the influence of each on measles importation events, as well as leverage reported imported measles data to strengthen our approach via cross-validation. Reported imported case data also allowed us to make and validate predictions regarding specific case counts. As a result, we developed a more detailed picture of imported measles dynamics in the US than might be possible without reported case data and facilitated an understanding of its drivers that might help to inform mitigation strategies and interventions such as targeted travel screenings.

Although our analysis yielded predictions that appear to explain a moderate degree of variation in imported measles cases across states and years, there are limitations of our analysis that should be acknowledged. First, our travel data were limited to air passenger travel data only and did not include potential instances of importation by land or water. Measles outbreaks on cruise ships could be one overlooked source of importation [28,29]. This data constraint also affects which potential importers can be accounted for, as international air travel is expensive. To the extent that individuals with reduced access to vaccines are less likely to travel internationally, the usefulness of measles incidence data for predicting imported cases could be limited. Second, the quality of the international measles data used is limited by the strength of reporting countries’ surveillance programs. Countries with poorer vaccination programs and more measles cases tend to have weaker surveillance systems, leading to underreporting. Though these countries are not necessarily highly connected to the US via air travel, such underreporting could nonetheless be present in model inputs. Third, our analysis was limited to imported case data that the CDC has made publicly available. Lack of data with which to validate estimates of country of origin limited our ability to draw conclusions about a causal relationship between model inputs and predicted case origins. Another study [30] was able to make use of more detailed information about imported case origin and month of detection held internally by the US CDC, which enabled monthly predictions of imported case magnitude and origin that we were unable to produce. Although it would be ideal to perform our analysis and share results in real time, the data necessary to implement such an analysis are not readily available. Other recent studies have overcome this obstacle by leveraging historical surveillance data, along with air travel data and vaccination exemption rates, to generate predictions of measles outbreak risk in the US [31].

Although we observed a strong association between model predictions and imported cases, other factors might be influencing these patterns, leading to unexplained variation and/or mis-attributed explained variation. The performance of our population only inflow scenario suggests that population plays a role in imported case dynamics, and it is likely that air traffic routes also drive international measles incidence patterns in some way. International vaccination efforts, not considered in this study, are another potential confounding factor present in the data; supplemental measles immunization programs have produced drastic reductions in measles burden around the world [32]. Though we did not explicitly control for alternative potential drivers of international measles transmission dynamics, we have attempted to limit the influence of domestic confounding factors by restricting our analysis to imported cases in the US.

At a time when measles cases are on the rise, the capability to predict the origin and magnitude of imported cases is a worthwhile endeavor that could potentially inform surveillance and prevention strategies. Although our models cannot determine the exact provenance of measles cases, the framework developed here provides a mechanism for explaining broader variation in patterns of imported measles cases in the US that could be validated with alternative data, such as sequence analysis. Given that the US CDC’s measles case definition states that “any case that cannot be proven as imported should be classified as indigenous” [9], some imported cases may be misclassified as indigenous. The capability that we advance here to generate quantitative predictions of imported cases could be useful in analyses that seek to more carefully disentangle importation and local transmission. The relationship between importation and local transmission can be complicated [33], and much remains to be understood about the interplay between measles importation patterns and local drivers of local measles outbreaks. Predictive models of imported cases such as ours stand to play an important role in better understanding these dynamics.

## 4. Materials and Methods

### 4.1. Data

The CDC in the United States publishes annual reports of notifiable disease cases, including state-level case counts for both imported and indigenous measles [34,35]. The CDC defines an imported case as one with a source outside the state, while an indigenous case must be unrelated to an imported case or occur more than two generations after a linked imported case [9]. Though the former definition allows for both international and out-of-state origin cases to be designated as imported, there was insufficient detail present in the data to separate the two. Because the majority of recent reported measles outbreaks in the US have been specifically associated with international importation events, we made the simplifying assumption that all imported cases had an international origin [36,37]. As such, we included all reported imported and reported indigenous measles cases from 2011 to 2016 in this study.

The WHO provides open access to monthly surveillance reports for measles in member nations [38]. We aggregated these data on an annual basis and used them in conjunction with United Nations (UN) annual population estimates to calculate yearly per capita measles incidence for member nations [39]. We also considered an alternative set of assumptions about international surveillance data that seek to correct for differential underreporting by country based on modeling [40,41,42,43,44], though data are available only by region beginning in 2013.

We obtained air passenger data from the International Air Transport Association (IATA) detailing monthly total passenger loads for flights from foreign countries to US states for the years 2011–2016 [45]. We additionally considered an alternative set of assumptions about air passenger travel data based on open-access model predictions, to permit sharing of runnable code [24].

### 4.2. Model Description

For each year of the study, 2011–2016, we calculated an inflow variable representative of the movement of measles into a state, with New York and New Jersey combined due to their airports being shared extensively. We included all countries for which flight and incidence data were present, for a total of 189 countries and 49 states as well as Washington, D.C. Each state- and year-specific inflow was a summation of the per capita incidence in a country multiplied by the flux—i.e., the number of air passengers from that country to a given state—for all countries to that state in a given year, or
(1)$${\mathrm{inflow}}_{j,y}=\sum_{i}{\mathrm{flux}}_{\mathrm{ij}}\;\times \;{\mathrm{incidence}}_{i}$$
for all countries *i* connected to state *j* in year *y*. We performed a Poisson regression of imported measles cases, *I*, against this inflow variable such that
(2)$$I_{j,y}\;\sim \;\mathrm{Poisson}(\lambda \;\times \;{\mathrm{inflow}}_{j,y}),$$
across all states *j* and years *y*. Across all states and years, this implied that the likelihood of λ was
(3)$$\mathrm{likelihood}(\lambda )=\prod_{j,y}^{}\frac{\lambda^{I_{j,y}}e^{-\lambda }}{I_{j,y}\text{!}},$$
which we used to estimate the value of λ by maximum likelihood using the optimize function in R [46]. 

### 4.3. Analyses

To evaluate model performance, we used Pearson correlation, coverage probability of the 95% prediction interval, and area under the curve of the receiver operating characteristic curve (AUC). Correlation and coverage probability allowed us to assess the model’s ability to account for variation in case numbers, whereas AUC allowed us to measure the model’s performance as a classifier of whether states experienced imported cases in a given year. We used the pROC package to compare and analyze ROC curves [47]. To quantify uncertainty for correlation and AUC, we sampled random subsets of 80% of the data 5000 times, repeated the regression analysis, and recalculated these statistics based on the model’s predictions of the remaining 20% of the data. Resampling was conducted on the inflow variables for all state–year combinations considered in the study. This also served as a form of cross-validation, given that it quantified model performance on data withheld from model fitting.

### 4.4. Alternative Inflow Variables

To evaluate the contribution each data type made to predictive performance, we repeated the regression analysis using modified inflow variables that replaced each of the two data types with averaged values. The averaged values of international incidence were defined as average annual incidence weighted by total outgoing passenger counts, or
(4)$${\mathrm{inflow}}_{j,y}\;=\;\sum_{i}{\mathrm{flux}}_{\mathrm{ij}}\;\times \;{\left(\frac{\sum_{i}{\mathrm{incidence}}_{i}\;\times \;\sum_{j}{\mathrm{flux}}_{\mathrm{ij}}}{\sum_{i}\sum_{j}{\mathrm{flux}}_{\mathrm{ij}}}\right)}_{y},$$
such that the inflow variable for state *j* and year *y* was a summation of the product of each country–state flux and a single average incidence value for year *y*. The averaged passenger flow values were defined as the product of the total number of passengers into state *j* and the proportion of passengers from country *i* to all US states, for year *y*,
(5)$${\mathrm{inflow}}_{j,y}\;=\;\sum_{i}\left(\sum_{i}{\mathrm{flux}}_{\mathrm{ij}}\;\times \;\frac{\sum_{j}{\mathrm{flux}}_{\mathrm{ij}}}{\sum_{i}\sum_{j}{\mathrm{flux}}_{\mathrm{ij}}}{\;\times \;\mathrm{incidence}}_{i}\right).$$

We also considered a null inflow value in which inflow was replaced by state population, or
(6)$${\mathrm{inflow}}_{j,y}{\;=\;\mathrm{population}}_{j,y}$$
[48]. Each alternative inflow variable was applied to Equation (2). This resulted in a total of six inflow scenarios, each a unique combination of flux and incidence data inputs (Table 3).

## Figures and Tables

**Figure 1 pathogens-10-00155-f001:**
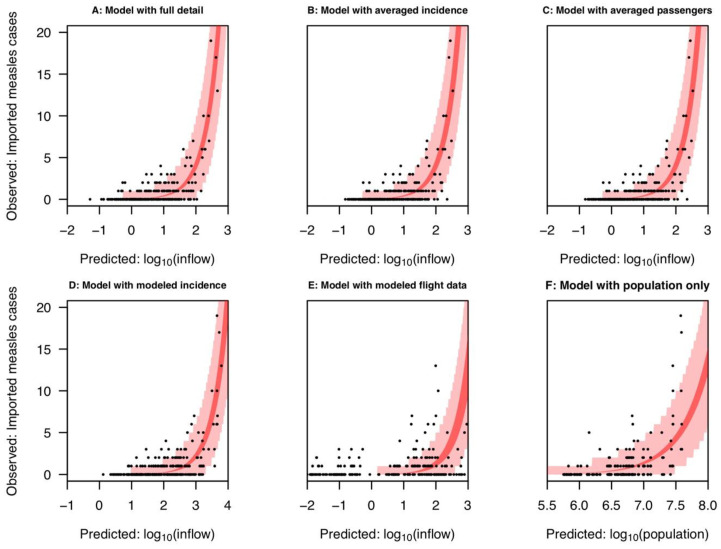
Observed imported measles cases accounted for by the fitted models (**A**) with full detail, (**B**) with averaged incidence, (**C**) with averaged passengers, (**D**) with modeled incidence, (**E**) with modeled flight data, and (**F**) with population only. Black dots represent imported measles cases by state and year. The dark red band indicates the 95% confidence interval of the fitted relationship, while the light red band indicates the 95% prediction interval of the data.

**Figure 2 pathogens-10-00155-f002:**
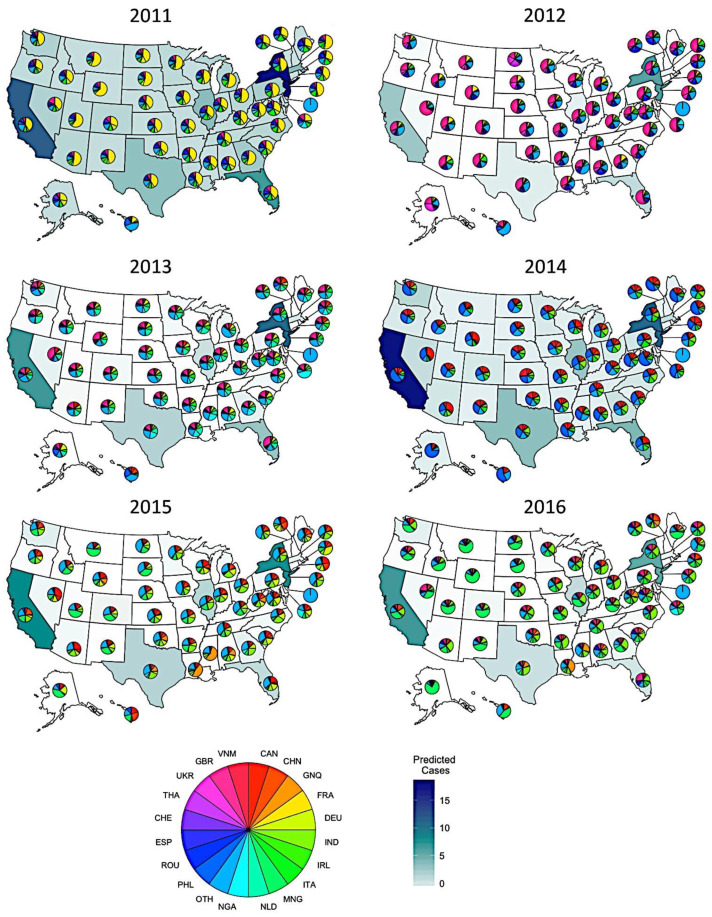
Predicted origins and magnitudes of measles importations into US states 2011–2016. Pie chart coloring indicates likely country of origin, while state fill color denotes magnitude of importation. As previously noted, New Jersey is grouped with New York, while Delaware has no major airport and is therefore marked as “Other” throughout. Countries shown on the pie charts are Canada (CAN), China (CHN), Equatorial Guinea (GNQ), France (FRA), Germany (DEU), India (IND), Ireland (IRL), Italy (ITA), Mongolia (MNG), Netherlands (NLD), Nigeria (NGA), Philippines (PHL), Romania (ROU), Spain (ESP), Switzerland (CHE), Thailand (THA), Ukraine (UKR), United Kingdom (GBR), Vietnam (VNM), and Other (OTH), which encompasses all other countries that were infrequently predicted as the origins of measles importations.

**Table 1 pathogens-10-00155-t001:** Summary statistics including the fitted parameter λ, correlation between observed and predicted imported cases, and area under the receiver operating characteristic curve.

Inflow Variable	λ	Correlation	AUC ^1^
Full detail	0.041	0.84	0.78
Averaged incidence	0.041	0.76	0.79
Averaged air travel	0.041	0.76	0.79
Modeled incidence	0.0022	0.77	0.79
Modeled flight data	0.012	0.61	0.67
Population only	4.32 × 10^−5^	0.66	0.58

^1^ Area Under the ROC Curve.

**Table 2 pathogens-10-00155-t002:** Cross-validated summary statistics with uncertainty, including the fitted parameter λ, correlation between observed and predicted imported cases, and area under the receiver operating characteristic curve (full distributions are shown in Appendix A).

Inflow Variable	Cross-Validated λ (Mean ± SD ^1^)	Cross-Validated Correlation (Mean ± SD)	Cross-Validated AUC (Mean ± SD)
Full detail	0.041 ± 0.002	0.78 ± 0.2	0.78 ± 0.06
Averaged incidence	0.041 ± 0.002	0.71 ± 0.2	0.79 ± 0.05
Averaged air travel	0.041 ± 0.002	0.76 ± 0.2	0.79 ± 0.05
Modeled incidence	0.0022 ± 1 × 10^−4^	0.77 ± 0.2	0.79 ± 0.06
Modeled flight data	0.012 ± 9.3 × 10^−4^	0.61 ± 0.3	0.67 ± 0.07
Population only	4.32 × 10^−5^	0.66 ± 0.1	0.58 ± 0.03

^1^ Standard Deviation.

**Table 3 pathogens-10-00155-t003:** Flux and incidence data inputs to Equation (1) for each inflow scenario.

Inflow Variable	Flux	Incidence
Full detail	Complete flight data	Complete incidence data
Averaged incidence	Complete flight data	Averaged incidence data
Averaged air travel	Averaged flight data	Complete incidence data
Modeled incidence	Complete flight data	Alternative incidence data
Modeled flight data	Alternative flight data	Complete incidence data
Population only	No flight data	No incidence data

## Data Availability

The publicly available datasets used in this study are cited throughout.

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
