# Peer review of "Air Passenger Travel and International Surveillance Data Predict Spatiotemporal Variation in Measles Importations to the United States"

_pathogens, 2021, doi:10.3390/pathogens10020155_

Round 1

Reviewer 1 Report

Abstract: I think there should be a summary sentence or two about how this study is important for predicting and understanding the dynamics of measles outbreaks in the future. 

Figure 1: There is no dark red band or light red band in the figure panels. 

Figure 2: Please list the countries for each abbreviation.

Reviewer 2 Report

This is an interesting and well written ms that illustrates how (meales) viruses are the great benefitters of globalization and international travel. Analyses appear to be well done and formulations are precise and careful.

Perhaps the only two suggestion I can make are

1.to more sharply define (early on) what an imported case is. Does it also include US citizens/residents who acquired infection abroad during, for example, a vacation?  

2. To explore (in a few sentences?) the impact of control measures such as requiring measles vaccination for international travelers below a certain age.

Reviewer 3 Report

General Comments

The authors provide an important and unique study on the impacts of newly introduced infected people with measles via air travel in the United States. This topic is of critical public health importance in the United States where many preventable diseases continue to increase. The authors evaluate the influx of new measles cases through various statistical approaches (i.e. regression models, correlation tests, are under the curve statistics, and resampling validation techniques. The study is overall very well written. Additional and organization of the methods would greatly enhance this manuscript.  

Detailed Comments

  1. The last paragraph read more like a methods section and less of a statement of the study design. It would be useful if the authors could clearly define their study aims with respect to measles in the US.
  2. This paper needs to have a methods section to discus source of data collection (both in terms of time and place), model choice(s), and any other potential confounders addressed for.
  3. The methods section needs to be moved to after the introduction. The methods should have sub-sections that clearly define the methods used for 2.1-2.5.
  4. The methods should clearly define variables of analysis per statistical model and if any approaches were used to control for confounding, clustering, etc.
  5. In terms of spatial epidemiology, the authors did to explicitly define the scale of space and time used in the statistical models.
  6. “A similar AUC of 0.78 ± 0.06” reads unclear. Please specify with SD or SE.
  7. “repetitive 87 resampling and cross-validation” – this needs to be better defined in the methods section. What are the time periods of resampling and how was the number of replicates defined?
  8. Figure 1 references a red band that is not present.
  9. The following predictors: “Full detail, Averaged incidence, Average air travel, Modeled incidence, Modeled flight data, and Population only” are not discussed previously. These need to be clearly defined in the methods.
  10. Table 1. The caption heading does not need to be centered.
  11. The discussion could be strengthened by discussing other major spatial and temporal factors that impact the spread of measles. For example, co-circulation of childhood diseases (see Noori and Rohani 2019; Philosophical Transactions of the Royal Society B) or network analysis.
